# In situ microphysical characterization of low-level clouds in the Finnish sub-Arctic site, extensive dataset.

Konstantinos Matthaios Doulgeris[1], Heikki Lihavainen[1,3], Anti-Pekka Hyvärinen[1], Veli-Matti Kerminen[2] and David Brus[1]

[1]Finnish Meteorological Institute, Erik Palménin aukio 1, P.O. Box 503, FIN-00100 Helsinki, Finland
[2]Institute for Atmospheric and Earth System Research/Physics, Faculty of Science, University of Helsinki, Helsinki, Finland
[3] Svalbard Integrated Arctic Earth Observing System (SIOS), SIOS Knowledge Centre, Svalbard Science Centre, P.O. Box 156, N-9171 Longyearbyen, Norway

*Correspondence to*: K.D (Konstantinos.doulgeris@fmi.fi)

**Abstract.** Continuous, semi-long-term, ground based in situ cloud measurements were conducted during eight Pallas Cloud Experiments (PaCE) held in autumns between 2004 and 2019. Those campaigns were carried out in the Finnish sub-Arctic region at the Sammaltunturi station (67º58´24´´N, 24º06´58´´E; 560 m MSL), the part of Pallas Atmosphere – Ecosystem Supersite and Global Atmosphere Watch (GAW) program. Two cloud spectrometer ground setups and a weather station were installed on the roof of the station to measure in situ cloud properties and several meteorological variables. Thus, the obtained dataset include the size distribution of cloud droplets as a measured cloud parameter along with the air temperature, dew point temperature, humidity, pressure, horizontal wind speed and direction, (global solar) sun radiation and visibility at the station. Additionally, the number concentration, effective diameter, median volume diameter and liquid water content from each instrument were derived. The presented data set provide a insight into microphysics of low-level clouds in sub-Arctic conditions over a wide range of temperatures (-25.8 to 8.8 ºC). The data are available in the FMI open data repository for each campaign and each cloud spectrometer ground setup individually: https://doi.org/10.23728/FMI-B2SHARE.988739D21B824C709084E88ED6C6D54B (Doulgeris et al.,2021).

## 1. Introduction

Clouds are considered as a major component of both the climate system and the hydrological cycle. Nevertheless, our level of understanding of the fundamental details of the cloud microphysical processes is still very limited (Boucher et al., 2013). To gain a deeper knowledge of the formation and development of the clouds, more in situ measurements are needed (Morrison et al., 2019). In addition, a correct representation of cloud microphysics in general circulation models for numerical weather and climate prediction is of great importance (Guichard and Couvrex, 2017; Morrison et al. 2020). Despite the fact that cloud processes can now be studied with much more confidence (Bony et al., 2015), representing the formation and evolution of cloud droplets and the effects of aerosols on clouds at various meteorological conditions remains a challenge (Grabowski et al., 2019). The number concentration and size distribution of cloud droplets are considered as key parameters for a quantitative microphysical description of clouds (e.g., Rosenfeld and Ulbrich, 2003; Komppula et al., 2005; Lihavainen et al., 2008, Pruppacher and Klett, 2010, Chang et al., 2019), and are connected with the cloud lifetime and radiative effects as well as precipitation (e.g., Albrecht 1989; Devenish et al., 2012, McFarquhar et al., 2020).

Three general approaches were used in previous studies of cloud microphysical properties: in situ sampling through airborne measurements by aircrafts (e.g. Heymsfield et al., 2011; Craig et al., 2014; Petäjä et al., 2016; Nguyen et al., 2021) and recently, by Unmanned Aerial Systems (UASs) (e.g. Girdwood et al., 2020; Brus et al., 2021; Harrison et al., 2021); in situ sampling by using laboratory cloud chambers (e.g. Möhler et al., 2003; Stratmann et al., 2004; Nichman et al., 2017; Doulgeris et al., 2018) and in situ ground based measurements (e.g. Guyot et al., 2015; Lloyed et al., 2015; Lowenthal et al., 2019; Doulgeris et al., 2020). In situ airborne and ground measurements (Wandinger et al., 2018) using cloud spectrometers are considered fundamental as they offer instrumental access to individual hydrometeors within a sampling volume. Unfortunately, each of the aforementioned approaches has inherent limitations.

Dataset that have been obtained from measurements in sub-Arctic clouds are significant as cloud processes are of high value since cloud processes are considered as an important component of climate change in the Arctic region (Wendisch et al., 2019). Pallas Cloud Experiments (PaCE) took place in the Finnish sub-Arctic. The main objective during PaCE was to study low-level clouds and their microphysical properties in a background sub-arctic environment. In this work, we present a unique dataset of ground in situ cloud measurements along with several meteorological variables collected at the Sammaltunturi station in eight autumn campaigns conducted between 2004 and 2019. This data set can be used in studies of cloud microphysics, climate change in sub-Arctic and performance evaluation and improvement of existing models, in particular at higher altitudes. In the next session, we provide a description of the sampling location, instrumentation, and the measurement methodology we used for sampling, data processing, and quality control.

## 2. Methods

### 2.1 Measurement site and PaCE campaigns overview

The Sammaltunturi station (67°58´24´´N, 24°06´58´´E) is hosted by the Finnish Meteorological Institute (FMI) and is located on a top of an arctic fjell (560 m above MSL) in the Finnish sub-Arctic region inside the Pallas–Yllästunturi National Park (Fig. 1). The Pallas area is located around 180 km above the Arctic circle, and it has no significant local or regional air pollution sources. Thus, the Sammaltunturi station provides an excellent location for the monitoring of background air composition in northern Europe. The station is about 100 m above the tree line and the vegetation around it consists mainly of low vascular plants, mosses, and lichen. There is a long history of atmospheric data collection in the area (see Lohila et al. 2015). Monitoring activities of atmospheric composition at Sammaltunturi started in 1991 in a building that originally served the Finnish Broadcasting Company. The new station (102 m$^2$) opened in July 2001. Since 1994, Sammaltunturi has been established as a node of the Pallas–Sodankylä supersite that contributes to the GAW program of the World Meteorological Organization. The site was described in detail in Hatakka et al. 2003. The main research measurements focus on greenhouse gas concentration, climate effects of atmospheric aerosols, aerosol cloud interaction and air quality (e.g., Komppula et al., 2005; Lihavainen et al., 2008; Asmi et al., 2011; Backman et al., 2017; Doulgeris et al., 2020). The predominant origin of air masses arriving at Sammaltunturi is from the Arctic (Asmi et al., 2011).

The main motivation to perform in situ cloud measurements at the Sammaltunturi was that the station was occasionally immersed in a cloud. Based on analytical data the most suitable time of the year for in situ cloud measurements was autumn when the horizontal visibility drops below 1 km around 40 % of the time (Hatakka et al. 2003). Once the preferable time of the year was identified, we started to conduct ground-based in situ measurements and study cloud formation. The "Pallas Cloud Experiments" were, usually, 6-8 weeks long and lasted approximately from the beginning of September until the end of November, occasionally extended to the beginning of December. The first attempt of measuring in situ cloud properties was made in 2004 using the forward-scattering spectrometer probe (FSSP-100) ground setup that was the only available cloud spectrometer at that time. The next campaigns, in 2005 and 2009, were done using the same instrument setup (Lihavainen et al., 2008). Later, in 2011 the cloud, aerosol and precipitation spectrometer (CAPS) ground setup was added. In January and February 2012, it was tested for the first time for two short periods during winter at the Sammaltunturi site. In 2012, 2013 and 2015 both instruments were installed and used during PaCE (Doulgeris et al., 2020). In 2017 and 2019, only CAPS was used (Girdwood et al., 2020). An overview of each year's campaign duration and the cloud spectrometer ground setups' availability is presented in figure 2. Instruments that were used for measuring the meteorological variables and the solar radiation were operating continuously during all PaCE years. The instrumentation used during PaCE campaigns is described in detail in the following section.

### 2.2 Instrumentation

In order to monitor meteorological variables, the station was equipped with an automatic weather station (Milos 500, Vaisala Inc.). A weather sensor (model FD12P, Vaisala Inc.) was used for measuring the horizontal visibility; the Vaisala HUMICAP was used for measuring the relative humidity; BAROCAP sensors were used for measuring the barometric pressure and PT100 sensors were used to measure temperature at 570 m. Global radiation and photosynthetically active radiation were measured with a pyranometer and a photovoltaic detector, respectively. Additionally, the wind speed was measured with a heated cup and the wind direction with a heated wind vane. All the above meteorological variables were saved as one-minute averages. A detailed description of the weather sensors can be found in Hatakka et al., (2003).

In order to conduct in situ cloud ground-based measurements, we deployed two instruments. The cloud, aerosol and precipitation spectrometer (CAPS) and the forward-scattering spectrometer probe (FSSP-100), (Droplet Measurement Technologies (DMT); Boulder, CO, USA) (Fig. 3). The FSSP (model SPP-100, DMT) was originally manufactured by Particle Measuring Systems (PMS Inc., Boulder CO, USA). Both instruments were originally developed for airborne measurements but modified as ground setups by the manufacturer (DMT, USA). They were installed on the rooftop of the Sammaltunturi station. The CAPS was fixed and heading always to the main wind direction of the station southwest, ~225°, while the FSSP-100 was installed on a rotating platform to continuously face the wind. The CAPS had a total height of 0.6 m above the roof where it was installed and a height of 4.5 m from the ground. FSSP had a total height of 0.6 m above the roof where it was installed and a height of 5.5 m from the ground. The two setups had a horizontal distance of ~10 m and vertical distance of ~1 meter between them. From 2004 until 2012 a flow laminator was used inside the FSSP inlet (Lihavainen et al. 2008). However, the flow laminator was often blocked by freezing or supercooled cloud droplets at sub-zero temperatures and for this reason it was cleaned every hour if occurrence of subcooled water was detected. The laminator blockage was evident both during everyday instrument inspection and from the raw data. Only data cleaned of this artefact were used in the FSSP data set. However, even without placing the laminator, the Reynolds number indicated that the flow inside the inlet was still laminar. As a result, in 2012 we decided that the laminator would not be used in the FSSP setup anymore. Thus, the amount of data after 2012 were more extensive and the number of cases when the FSSP would have been blocked was significantly reduced. A detailed description of both ground setups and the methodology we used for obtaining the ground-based cloud microphysical properties with in situ method was documented in Doulgeris et al., (2020). Only a short overview is given here.

The CAPS has been widely used in airborne measurements of the microphysical properties in clouds (e.g., Baumgardner, 2001; Baumgardner et al., 2011; DMT Manual; 2011 Lachlan-Cope et al. 2016). The CAPS probe includes three instruments; the cloud and aerosol spectrometer (CAS) which measures smaller particles, the cloud imaging probe (CIP) and the hot -wire liquid water content (LWC$_{hw}$) sensor. For the ground setup we deployed, the hot-wire LWC faced difficulties to operate in such extreme conditions; after operating in supercooled liquid clouds (even for a short time) the sensor was accreting ice. In addition, the lifetime of the sensor is limited and significantly shorter than the duration of the campaign. The FSSP-100 was widely used for measuring droplet size distribution (e.g., Brenquier, 1989; Lihavainen et al.2008; Lloyd et.al. 2015; Doulgeris et al. 2020). CAS and FSSP-100 derive the size of the particle from the intensity of the scattered light, using the Mie theory (Mie, 1908). Furthermore, backscatter optics measure light intensity in the 168 to 176° range. This allows the determination of the real component of a particle's refractive index for spherical particles. The CIP is a single particle optical array probe. Its design is based on optical measurement techniques whereby single particles pass through a collimated laser beam and their shadow is projected onto a linear array of 64 photodetectors. The count of the particle is dependent on the change in the light intensity of each diode.

All the instruments were calibrated before and after each campaign. Until 2011, we relied on the manufacturer calibration that was done at DMT. After 2011, we also started to perform calibration at the FMI, on top of manufacturer calibration, to ensure the quality of the collected data. For the calibration of the CAS and FSSP-100, glass beads in the diameter size range 2-40 µm and polystyrene latex sphere (PSL) standards in the diameter size range 0.74 - 2 µm were used. Cloud spectrometers (in our case CAS and FSSP-100) are calibrated for size measurements but not for number concentration measurements. The instruments faced extreme conditions during the whole campaign, in terms of frequent changes in wind direction, wind speed and sub-zero temperatures.

Despite the calibration procedures we should always keep in mind that extreme meteorological conditions could possibly lead to unexpected performance. To calibrate the CIP, a spinning glass disk with opaque dots of known size was used.

The CAPS ground setup included a high-flow pump (Baldor, Reliance, USA) which was working as an aspiration system. The aspiration system was made and provided by the manufacturer (DMT). A custom aspiration system with high flow ventilator was also made by the manufacturer (PMS) and employed through FSSP-100 inlet to ensure constant flow through it. A digital thermo-anemometer (model 471, Dwyer Inc.) was used in each campaign for checks of daily cloud spectrometers' air speed. The FSSP air speed inside the inlet was calculated from the measured airspeed in front of the inlet, except in 2004 and 2005 when the air speed was calculated with measured volume flow rate through the inlet. A necking inside the inlet led the flow from inner diameter 3.8 cm to 2.0 cm. Both spectrometers were equipped with anti-ice systems as they were modified by the manufacturers (DMT for CAPS and PMS for FSSP-100) for ground-based use. Despite the existing anti-ice features, due to the subzero temperatures that they were facing, snow or ice could accrete and affect the airspeed inside the probe inlets. For this reason, to ensure the proper operation of the instruments, they were inspected and cleaned twice per day, every morning and evening (approximately every 12 hours).

The ground-based in situ cloud measurements provided the cloud and precipitation size distribution. The PADS 2.5.6 software that was used for the data acquisition of CAPS measurements (DMT Manual, 2009), provided the number concentration ($N_c$, cm$^{-3}$), liquid water content (LWC, g cm$^{-3}$), median volume diameter (MVD, µm) and effective diameter, (ED, µm). For the FSSP-100, $N_c$, LWC, MVD and ED were also derived using the same equations (Doulgeris et al., 2019), since we have used an older software for data acquisition (PACS 2.2, DMT).

The major sources of uncertainties of the cloud spectrometers can be coincidence, dead time losses and changing velocity ratio (Guyot et al.,2015). The uncertainty of estimation of sizing at the cloud spectrometers was as 20% and of the number concentration was as 16% (Baumgardner, 1983; Dye and Baumgardner, 1984; Baumgardner et al., 2017). According to Lance (2012), it was observed that for CAS at ambient droplet concentrations of 500 cm$^{-3}$ there was 27 % undercounting and a 20 – 30 % oversizing bias. In our case, during PaCE campaigns the droplet number concentration values we monitored were in the majority of cases less than 300 cm$^{-3}$. These number concentration values lead us not to take coincidence, dead-time losses, and VAR uncertainties into consideration in this analysis. LWC has a significant uncertainty of 40% (DMT manual, 2009). The FSSP derived ED and LWC had an uncertainty of 3 µm and 30 % in mixed-phase clouds (Febvre et al. ,2012). An overview of the instrumentation and their operational characteristics we used for cloud measurements are summarized in Table 1.

## 3. Overview of data set and quality control description

The current dataset contains only in-cloud measurements when the station was immersed in a cloud. Data from each cloud probe and the weather station were quality controlled and unified in a common format for release and further analysis. The presence of a cloud at the station was identified with three different factors. First, we checked the droplet size distribution measured in both the cloud spectrometers. This was the main parameter to consider that the station was inside a cloud. Then, to confirm this assumption, we crosschecked the droplets counts with two meteorological variables; the relative humidity at the measurement site which was expected to be ~100 % and the horizontal visibility which should be less than 1 km, when the Sammaltunturi station is in the cloud. In case that one of the factors was not fulfilled, a final inspection was done visually using pictures recorded by an automatic weather camera installed on the roof of the station.

During PaCE 2004 and 2005 the sampling time of the FSSP-100 was 15 s. During PaCE 2009 the instrument was set to sample at 10s. From 2009 until 2019 the sampling time was set to sample each 1 s (1 Hz) for both instruments. PT100 sensor, Vaisala HUMICAP and BAROCAP sensors, the pyranometer, the heated cup and wind vane were also set to sample to 1 s. FD12P Vaisala weather sensor sampling time was 15 s. For every year, one-minute averages were calculated for each cloud spectrometer and each meteorological variable. As a result, we obtained the cloud droplet size distribution and several meteorological variables

for each minute and as derived parameters the $N_c$ (cm$^{-3}$), LWC (g cm$^{-3}$), MVD (µm) and ED (µm). All data sets were converted to NetCDF format. All times in this work are given in UTC time. Our dataset includes a separate NetCDF and .cvs file for each cloud spectrometer and for each year under the file name PACE.yyyy.cloud_spectrometer.nc and PACE.yyyy.cloud_spectrometer.cvs. (example names). For every file, the sampling area (mm$^2$) and the probe air speed (ms$^{-1}$) that was used to derive each parameter is provided. In addition, it includes the cleaned timeline data set of the following cloud properties and meteorological variables: Year (YYYY), day (DD), month (MM), hour (HH), min (MN), size bin lower limit, size bin higher limit, number concentration (cm$^{-3}$), liquid water content (g cm$^{-3}$), effective diameter (µm), median volume diameter (µm), the calculated d$N$/dlog$D$p (cm$^{-3}$) values in each bin, temperature at 570 meters (°C), dew point (°C), humidity at 570 meters (%), pressure (hPa), wind speed (m s$^{-1}$), horizontal wind direction (degrees), global solar radiation (Wm$^{-2}$), photosynthetically active radiation (µmol m$^{-2}$ s$^{-1}$) and the horizontal visibility (m). The derived cloud parameters (number concentration (cm$^{-3}$), liquid water content (g cm$^{-3}$), effective diameter (µm), median volume diameter (µm)) were not included in the CIP files. The number of cloud droplets per minute in CIP size range lead to statistically biased values and for this reason we decided to exclude them. The variables, naming abbreviations and units are summarized in Table 2.

The CAS contains 30 size bins with forward scattering upper bin size of 0.61, 0.68, 0.75, 0.82, 0.89, 0.96, 1.03, 1.1, 1.17, 1.25, 1.5, 2, 2.5, 3, 3.5, 4, 5, 6.5, 7.2, 7.9 10.2, 12.5, 15, 20, 25, 30, 35, 40, 45 and 50 µm and the CIP contains 62 size bins with bin size of 15, 30, 45, 60, 75, 90, 105, 120, 135, 150, 165, 180, 195, 210, 225, 240, 255, 270, 285, 300, 315, 330, 345, 360, 375, 390, 405, 420, 435, 450, 465, 480, 495, 510, 525, 540, 555, 570, 585, 600, 615, 630, 645, 660, 675, 690, 705, 720, 735, 750, 765, 780, 795, 810, 825, 840, 855, 870, 885, 900, 915 and 930 µm. For the FSSP-100 two different bin size ranges were used. During 2004 and 2005 the instrument was set up to use 30 size bins with forward scattering upper bin size of 3.0, 4.5, 6.0, 7.5, 9.0, 10.5, 12.0, 13.5, 15.0, 16.5, 18.0, 19.5, 21.0, 22.5, 24.0, 25.5, 27, 28.5, 30.0, 31.5, 33.0, 34.5, 36.0, 37.5, 39.0, 40.5, 42.0, 43.5, 45.0, 47.0. From 2009 until 2015, the FSSP was set up to use 40 size bins with forward scattering upper bin size of 1.2, 2.4, 3.5, 4.7, 5.9, 7.1, 8.2, 9.4, 10.6, 11.8, 12.9, 14.1, 15.3, 16.5, 17.6, 18.8, 20, 21.2, 22.3, 23.5, 24.7, 25.9, 27, 28.2, 29.4, 30.6, 31.7, 32.9, 34.1, 35.3, 36.4, 37.6, 38.8, 40, 41.1, 42.3, 43.5, 44.7, 45.8 and 47 µm.

Measurements of each year were inspected to ensure a good quality of the data set. First, the raw dataset was checked in order to eliminate and exclude from further analysis cases when one of the cloud probes was partially or fully blocked. Partially or fully blocked probes were also visible in raw data. To detect blocked probes, Nc was carefully investigated for the whole dataset. When a sudden decrease just before a sudden increase in droplet number concentration was occurring, we had a clear sign of probe inlet freezing. This behavior was observed due to the opening of the probe inlet becoming smaller (from the accumulation of snow/ ice) and resulted in a raised probe air speed. During data evaluation we considered that the probe air speed was constant. This abnormality in the Nc was happening due to the underestimation of the probe air speed. Then, we applied the suggested corrections due to limitations (Doulgeris et al., 2020) for the data analysis of the CAS and FSSP-100 ground setups. Doulgeris et al., 2020 demonstrated that the CAPS (that was fixed to one direction) showed significant sampling losses when it was not facing the wind direction since it was not sampling isokinetically. For this reason, the data that were obtained in the wind iso-axial conditions were considered to have the best quality. Thus, regarding CAPS, only the measurements when the instrument was facing the wind direction were included. FSSP-100 ground setup was always directed against the wind direction and as a result we provided measurements from all wind sectors. Missing data points were marked as -9999.9.

As it is shown in Fig.4, the observation hours after PaCE 2013 when the campaigns had longer duration are significantly higher. The amount of data in these years is excessive serving as an important source of information for Arctic studies. An overview of meteorological variables is presented for each campaign when the FSSP-100 and CAPS ground setups were operational. In Fig. 5, a statistical description of the temperature at 570 MSL for each campaign is illustrated. Each PaCE year the temperature trends and ranges were similar (around - 10.0 to 8 °C). In Fig. 6, we show the percentage of the data set for each year in which the Global solar radiation was higher than 0. It was used to estimate the amount of data collected in each campaign in day light. In addition, an overview of the microphysical derived cloud properties data from each campaign is presented. Thus, in

Fig.7, Fig.8 and Fig.9, the number concentration, the effective diameter, the medium volume diameter and the liquid water content are presented for each campaign and for the FSSP-100 and CAS ground setups, respectively. Number concentration averaged values were similar for every year of the measurements and reach scales around 100 cm$^{-3}$. However, there were some cloud cases during each campaign that number concentration had values around 300 cm$^{-3}$. The averaged ED and MVD values were ranging approximately from 10 to 20 µm. The liquid water content was less than 0.2 g cm$^{-3}$ in most cases.

## 4. Data availability

Each described dataset was collected by Finnish Meteorological Institute during PaCE campaigns and was published in the described form at FMI open data repository. All data set have undergone thorough quality control and false readings were eliminated. Dataset can be all found here: https://doi.org/10.23728/FMI-B2SHARE.988739D21B824C709084E88ED6C6D54B (Doulgeris et al.,2021). When the CIP was operational, we also collected the CIP images. However, we did not include the raw images in the data set for two reasons, First, there were in binary format. To read them, we used a proprietary image analysis software that was provided by DMT. Secondly, the upper limit of the open data repository is 10GB which was not enough to include the CIP raw images which were approximately 0,5 GB per case/day. However, RAW CIP images could be provided on demand by authors.

## 5. Code availability

Software developed to process and display the data from the cloud ground base spectrometers are not publicly available and leverages licensed data analysis software (MATLAB). This software contains intellectual property that is not meant for public dissemination.

## 6. Summary

In this study we produced and summarized dataset obtained from two cloud ground base spectrometers (CAPS and FSSP-100 ground setups) owned by the FMI during eight years of PaCE campaigns conducted during autumns from 2004 until 2019 along with several meteorological variables. PaCE campaigns took place in the Finnish sub-arctic region in a clear environment in temperatures that were usually below zero. In section 2, we describe the measuring site where PaCE campaigns took place and the cloud ground spectrometers setups that were used to obtain the cloud data along with the instrumentation that was used to monitor the weather conditions. In Section 3 an overview of the data set is presented.

These observations gathered in sub-arctic conditions are a unique source of in situ cloud measurements, which can contribute to the understanding of the cloud dynamics and formation in a sub-arctic environment in different meteorological conditions. Such semi long observations are difficult to obtain in similar environments due to current lack of instrumentation which would allow continuous unattended operation at temperature below 0 $^{\circ}$C. Cloud droplet spectrometers with surface installation had been identified as a potential method for continuous cloud in-situ measurements (Wandinger et al., 2018). Thus, due to the increased demand for long term continuous ground based in-situ cloud measurements, we provide a data set of in situ cloud measurements in a harsh sub arctic environment Each data set includes a combination of cloud microphysical properties along with several meteorological variables. Even though the data set includes measurements from eight campaigns, we would propose a case-by-case cloud investigation. Due to the inhomogeneity of the presented cloud cases, it is challenging to retrieve any trend that can be unambiguously connected to changes in the atmosphere. Also, the quality of dataset may differ for each campaign due to the different

amount of observations per year and operators' experience running the ground-based spectrometers through the years. In addition, each cloud case could be of different mass origin. We therefore discourage from any trend analysis based only on the presented data set. At least thorough back – trajectories analysis and subsequent segregation of dataset according to air mass origin is recommended. However, this was not an objective of this manuscript. The dataset in current form provides a helpful contribution to cloud microphysics processes on shorter timescales. Microphysical processes can strongly influence cloud-climate feedbacks in global climate models (Bodas – Salcedo et al., 2019). Furthermore, it can be used as complementary in model development. Representation of cloud microphysics is considered significant for large eddy simulation models (LES) (Morrison et al., 2020). There is a need for in situ cloud datasets due to two significant problems that the modeling community is facing; the representation of the population of the cloud and precipitation particles and the uncertainties due to fundamental gaps in knowledge of cloud physics (Morrison et al., 2020). In this dataset, the cloud size distribution was monitored in different stages of its evolution.

Appendix A: Abbreviations

| | |
|---|---|
| PaCE | Pallas Cloud Experiment |
| GAW | Global Atmosphere Watch |
| UAS | Unmanned Aerial System |
| FMI | Finnish Meteorological Institute |
| CAPS | Cloud, aerosol and precipitation spectrometer |
| CAS | Cloud and aerosol spectrometer |
| CIP | Cloud imaging probe |
| $LWC_{hw}$ | Hot -wire liquid water content sensor |
| FSSP -100 | Forward-scattering spectrometer probe |
| DMT | Droplet Measurement Technologies |
| PMS | Particle Measuring Systems |
| PSL | Polystyrene latex sphere |
| $N_c$ | Number concentration |
| LWC | Liquid water content |
| ED | Effective diameter |
| MVD | Median volume diameter |
| $T$ | Temperature at 570 MSL |
| $T_{DP}$ | Dew point temperature |
| RH | Relative Humidity at 570 MSL |
| $P$ | Pressure |
| $W_s$ | Wind speed |
| $W_{dir}$ | Wind direction |
| $S_{rad}$ | Global solar radiation |
| PAR | Photosynthetically active radiation |
| $V$ | Horizontal Visibility |

*Author contributions.* KD wrote the paper with contributions from all co-authors. HL planned and coordinated PaCE 2004, 2005, 2009. HL and DB planned and coordinated PaCE 2012 and 2013. KD and DB planned and coordinated PaCE 2015, 2017 and 2019. KD and DB processed, analyzed and quality controlled the data set. VMK and APH reviewed and edited the manuscript.

*Competing interests.* The authors declare no conflict of interest.

*Acknowledgements:* This work was supported by the Koneen Säätiö (grant no. 46-6817), the NordForsk (grant no. 26060), the Academy of Finland (grant no. 269095), the Academy of Finland Center of Excellence program (grant no. 307331), Academy of Finland Flagship funding (grant no. 337552), the Natural Environment Research Council (NERC (grant no. NE-L011514-1)). This
project has received funding from the European Union, Seventh Framework Programme (BACCHUS) (grant no. 603445)) and H2020 research and innovation program (ACTRIS-2, the European Research Infrastructure for the observation of Aerosol, Clouds, and Trace gases) (grant agreement no. 654109).

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

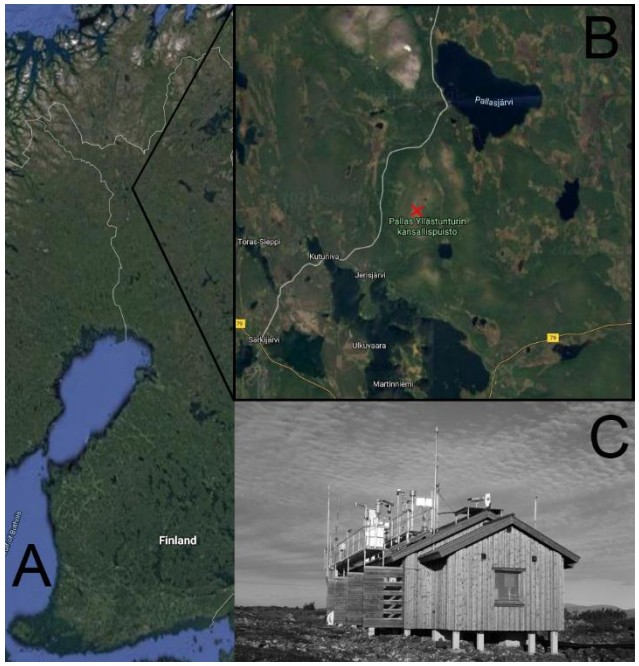


Figure 1. (a) Map of Finland showing the location of the field station, and (b) map of the wider Pallas area showing the location of the Sammaltunturi station (red cross). © Google Maps (c) The Sammaltunturi measuring station during PaCE.



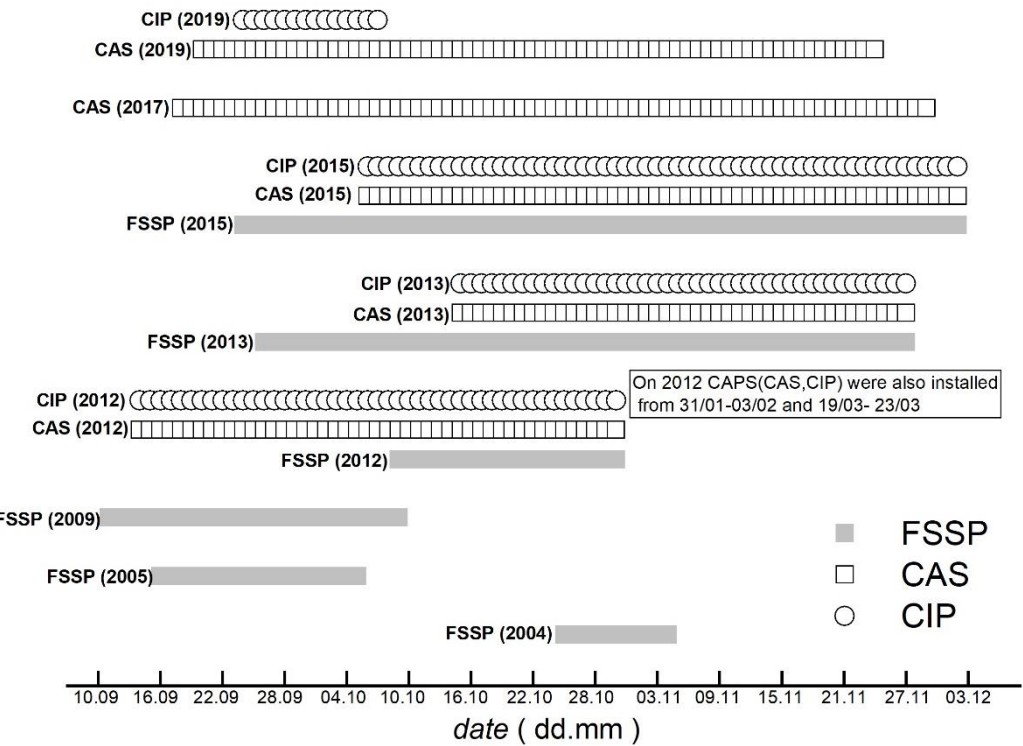

Figure 2. Cloud spectrometer ground setups' availability during PaCE is presented for each year.

**Table 1:** An overview of instrumentation and their operational characteristics provided by manufacturer.

| Instrument | Operating range | Number of bins | Sampling frequency | Air speed range | Accuracy | Uncertainties |
|---|---|---|---|---|---|---|
| *Cloud instruments* | | | | | | |
| *CAS, DMT* | 0.51 µm to 50 µm | 10, 20, 30, or 40 | 0.05 to 40 Hz | 10 - 200 ms$^{-1}$ | upper $N_c$ > 1,000 cm$^{-3}$ after corrections for coincidence that are about 25% at 800 and 30% at 1,000 particles/cm$^3$ Sizing accuracy: 20% | ambient $N_c$ of 500 cm$^{-3}$: 27% undercounting and 20%–30% oversizing bias Lance et al. (2012) *LWC*: 40% (DMT Manual) |
| *CIP*, DMT | 12.5 µm to 1.55 mm | 62 | 0.05 to 40 Hz | 10 - 300 ms$^{-1}$ | upper $N_c$ range up to 500 particles/cm$^3$ for a CIP with standard tips and arm width sizing accuracy: 1 $\mu$m | digitization uncertainty of approximately 61 size resolution that depends upon where the particle passes across the array Baumgardner et al. (2017) |
| *FSSP-100,* PMS | 0.5 µm to 47 µm | 15,30 or 40 | 0.05 to 40 Hz | | $N_c$ accuracy: 16% sizing accuracy: ±3 µm *LWC* accuracy: 30%–50% Baumgardner (1996) | derived *ED*: 3µm derived *LWC*: 30% Febvre et al. (2012) |

| *Meteorological instruments* | | | | |
|---|---|---|---|---|
| | Range | Resolution | Sensitivity | Accuracy |
| *PT100 sensor*, Vaisala | -70 – +180 (°C) | 0.01 (°C) | | ±0.1 (°C) |
| *HUMICAP sensor*, Vaisala | 0 – 100 (%) RH | <0.01 (%) RH | | ±0.8 (%) RH |
| *BAROCAP sensor*, Vaisala | 500 – 1000 (hPa) | 0.01 (hPa) | | ±0.15 (hPa) |
| *heated cup and wind vane*, Vaisala | 0.4 – 75 (ms$^{-1}$) 0 – 360° | 0.1 (ms$^{-1}$) 1° | | ±0.17 (ms$^{-1}$) ±3° |
| *Pyranometer*, Vaisala | 305 – 2000 (Wm$^{-1}$) | | 9– 15 (µV Wm$^{-2}$) | < ±20 Wm$^{-2}$at 1000 Wm$^{-2}$ |
| *FD12P*, Vaisala | 10 - 50000 (m) | | | ±10 %, 10 –10000 m ±20 %, 10000 –50000 m |


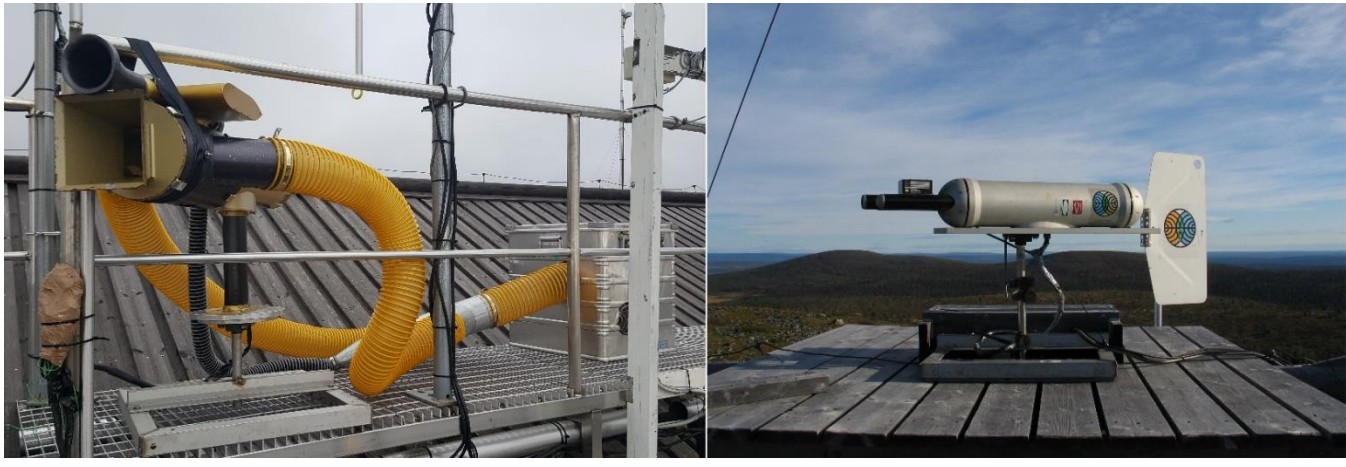

Figure 3. CAPS (left) and FSSP-100 (right) ground setups as installed on the roof of Sammaltunturi station.







**Table 2:** Cloud properties and meteorological variables along with abbreviations and units as they are included in each dataset.

| Variable name | Abbreviations | Units | Comments |
|---|---|---|---|
| *Cloud properties* | | | |
| Number concentration | $N_c$ | $cm^{-3}$ | derived parameter |
| Liquid water content | LWC | $g\ cm^{-3}$ | derived parameter |
| Effective diameter | ED | µm | derived parameter |
| Median volume diameter | MVD | µm | derived parameter |
| Size distribution | d$N$/dlog$D$p | $cm^{-3}\ µm^{-1}$ | calculated from min averages counts per bin |
| *Meteorological variables* | | | |
| Temperature at 570m | $T$ | ºC | PT100 sensor |
| Dew point temperature | $T_{DP}$ | ºC | |
| Relative Humidity at 570m | RH | % | Vaisala HUMICAP sensor |
| Pressure | $P$ | hPa | Vaisala BAROCAP sensor |
| Wind speed | $W_s$ | $ms^{-1}$ | measured with a heated cup |
| Wind direction | $W_{dir}$ | degrees | measured with a heated wind vane |
| Global solar radiation | $S_{rad}$ | $Wm^{-2}$ | Pyranometer |
| Photosynthetically active radiation | PAR | $µmol\ m^{-2}\ s^{-1}$ | Photovoltaic detector |
| Horizontal Visibility | $V$ | m | FD12P Vaisala weather station |

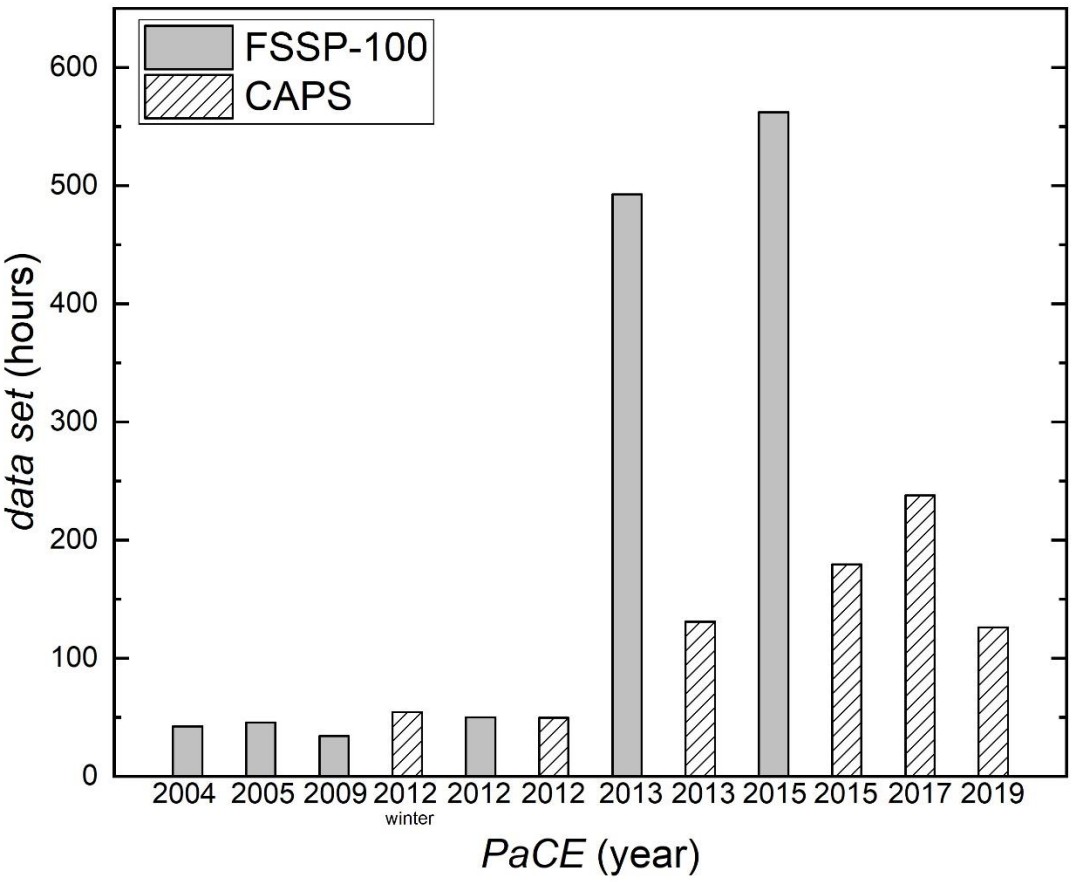


Figure 4. Hours of observation data collected for each PaCE campaign when the FSSP-100 and CAPS ground setups were operational.


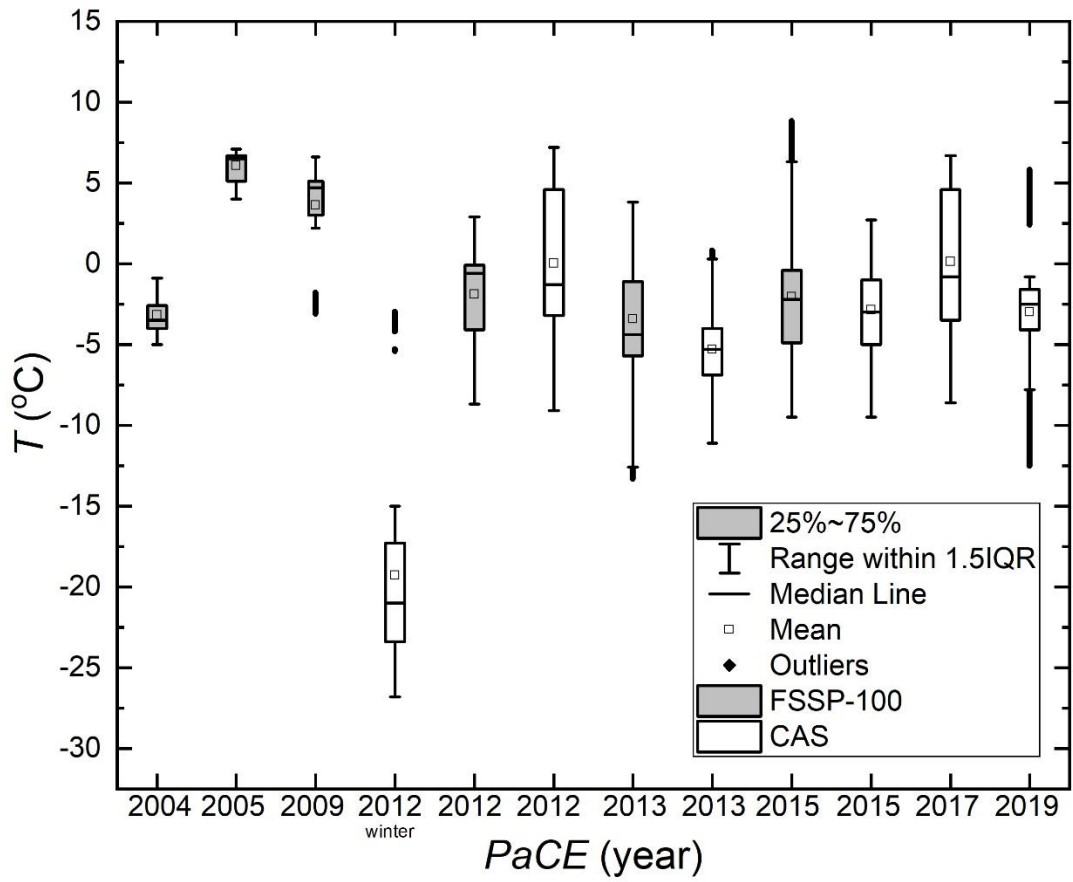

Figure 5. Statistical description of the temperature at 570m above MSL for each PaCE campaign when the FSSP-100 and CAS ground setups were operational.

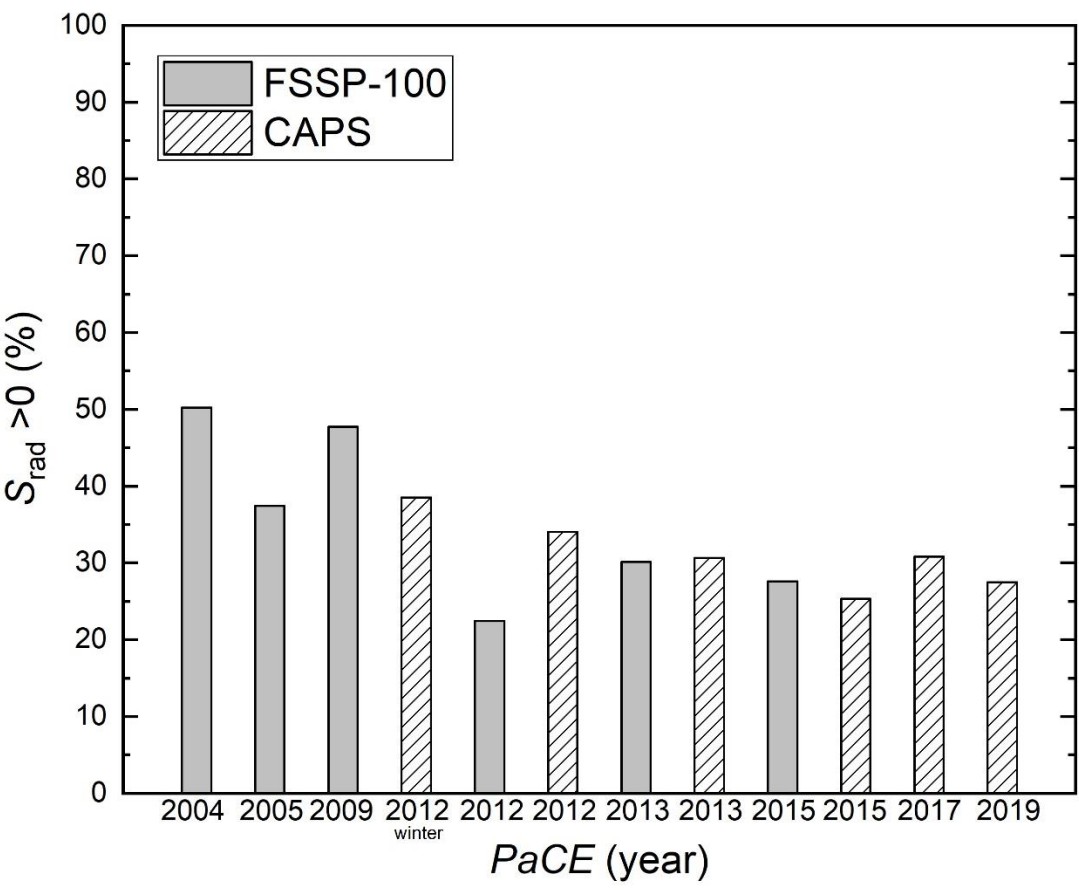


Figure 6. The percentage of the global solar radiation that was higher than 0 during each campaign when the FSSP-100 and CAS ground setups were operational.


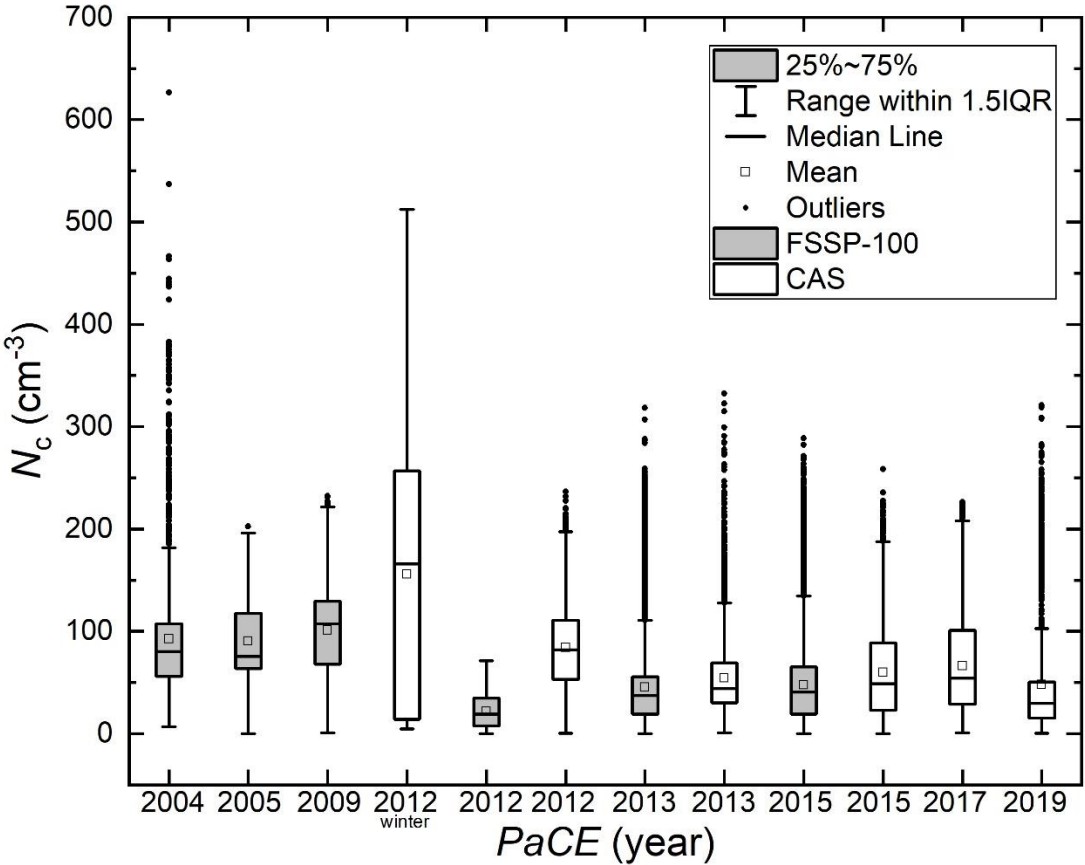

Figure 7. Statistical description of $N_c$ for each PACE campaign during the FSSP-100 and CAS ground setups were operational.

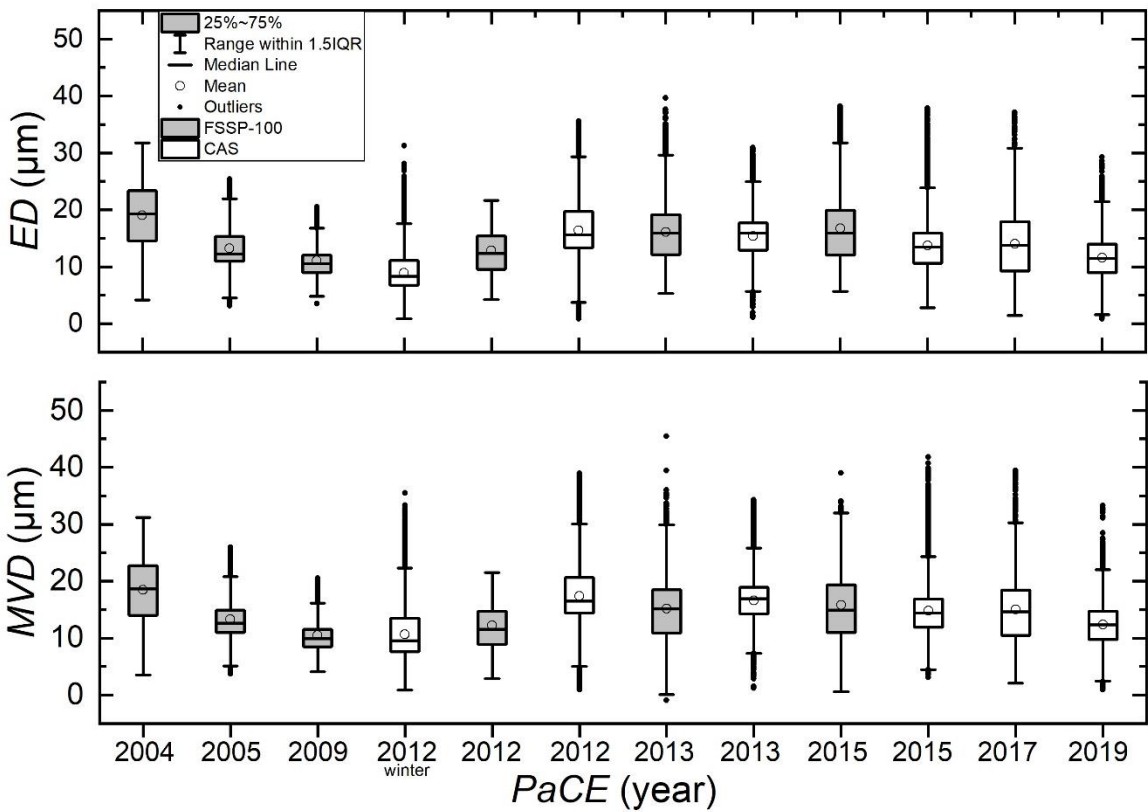

Figure 8. Statistical description of *ED* (upper panel) and *MVD* (lower panel) for each PACE campaign during the FSSP-100 and CAS ground setups were operational.


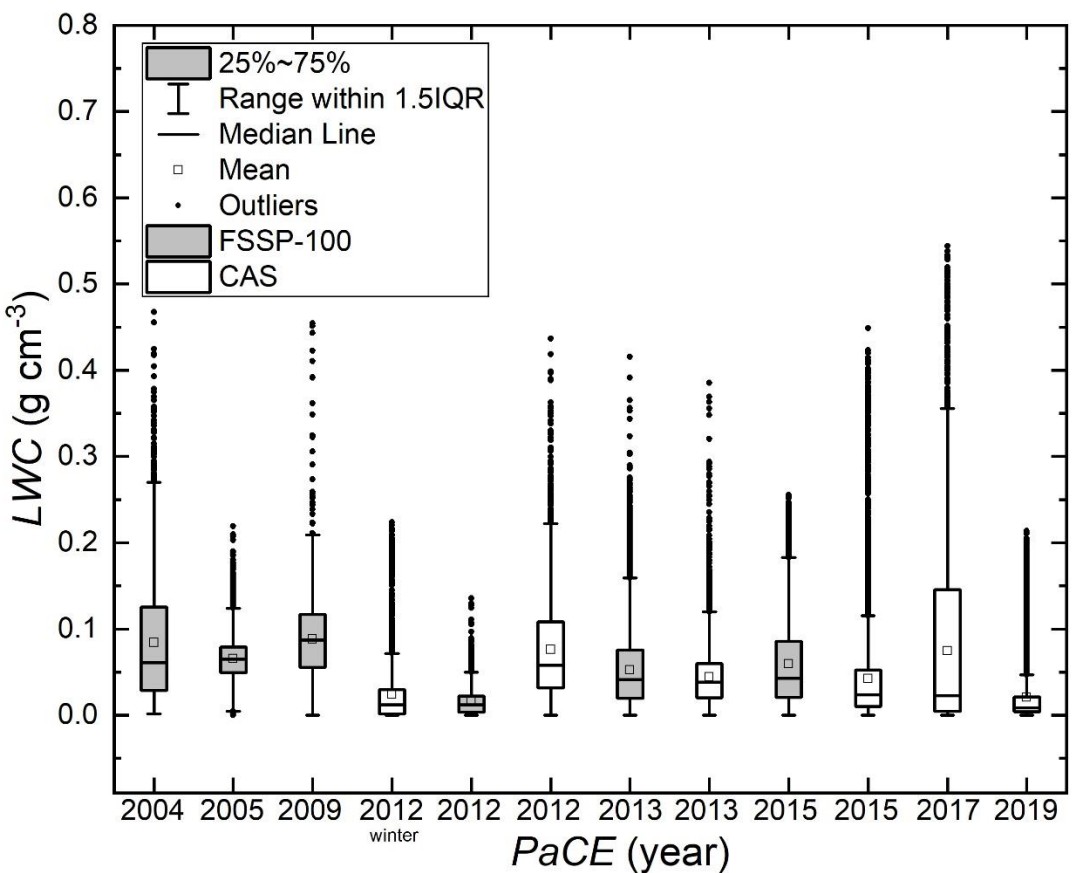

Figure 9. Statistical description of *LWC* for each PaCE campaign when the FSSP-100 and CAS ground setups were operational.




