# Peer review of "In situ microphysical characterization of low-level clouds in the Finnish sub-Arctic site, extensive dataset."

_Earth System Science Data, 2021_

## Author Comment (AC1)

We sincerely thank the reviewer #1 for carefully reading of our manuscript, for his review and constructive comments. We have reviewed the comments and have revised the manuscript accordingly. Our response is given in a point-by-point manner below. Reviewer comment (RC) Authors answer (AA).

*The authors produced and summarized data obtained from two cloud ground based spectrometers (CAPS and FSSP-100 ground setups) and accompanying meteorological instruments during eight years of cloud measurements in field campaigns conducted in the Finnish sub-arctic region, during autumns from 2004 until 2019.*

**Major comments:**

RC1:*The authors state multiple times that the provided datasets are significant and very important for cloud microphysics, but don't provide any evidence or specific examples of how their datasets can be used. Instead, the authors highlight that these data can't be used for trend analysis and that different campaigns shouldn't be combined into one analysis (L227-232). I'd highly recommend to include at least one specific example in this manuscript where these datasets are used.*

AA1: Text will be added in the revised manuscript (discussion section) to discuss and clarify our aim and the importance of the provided dataset. We want to contribute to the community by making available a large in situ cloud data set. Indeed, we don't recommend the dataset to be used for trend analysis due to the inhomogeneity of the presented cloud cases as it is challenging to retrieve any trend that can be unambiguously connected to changes in the atmosphere. However, Morrison et al. (2020) presented an extensive discussion regarding the challenge of modeling cloud and precipitation microphysics. There, they addressed two major problems of the modeling community; the representation of the population of the cloud and precipitation particles and the uncertainties due to fundamental gaps in knowledge of cloud physics. To investigate those gaps, there is a need for in situ cloud data sets. Also, Bodas – Salcedo et al. (2019) highlighted that the representation of microphysical processes could influence cloud climate feedbacks in global climate models. Our paper helps exactly towards these objectives as we monitored the cloud size distribution in different stages of its evolution (dataset includes cases where we follow the low-level cloud from its genesis until to precipitation) along with measuring several meteorological parameters. Thus, an investigation can be done on how different meteorological conditions could affect cloud formation and the cloud microphysics.

Both Morrison et al. 2020 and Bodas – Salcedo et al.2019 will be added to the reference list of the revised manuscript.

RC2:*The figures, particularly figures 4-9, are barely discussed in the main text.*

AA2: Text will be added in the revised manuscript to discuss the figures, especially figures 4-9. However, the decision of not discussing the figures in detail was taken into due to the general guidelines of the ESSD journal that" Articles in the data section may pertain to the planning, instrumentation, and execution of experiments or collection of data. Any interpretation of data is outside the scope of regular articles.". Yet another manuscript including analysis and discussing results from dataset presented here is in preparation.

RC3: *The grammar needs to be revised, I provided some recommendations in the minor comments below.*

AA3: Grammar will be revised.

**Minor comments:**

RC4: *I'd suggest to change the title to: In-situ microphysical characterization of low-level clouds in the Finnish sub-Arctic site, in the years 2004-2019.*

AA4: Manuscript title was changed to "*In-situ microphysical characterization of low-level clouds in the Finnish sub-Arctic site, extensive dataset.*" "site "was added according to reviewer suggestion. To avoid any misunderstandings and possible confusions about the content of this manuscript we kept the extensive dataset in the title. Also, we would like not include "years 2004 – 2019" to avoid any confusion since the campaigns were not organized every year.

RC5: *L15: cloud parameter along with the air temperature*

AA5: The above suggestion was accepted.

RC6: *L16: horizontal wind speed I suppose?*

AA6: The above suggestion was accepted.

RC7: *L18: remove: "includes cloud cases with temperature from"*

AA7: The above suggestion was accepted.

RC8: *L19: The data are available in the FMI...*

AA8: The above correction was applied.

RC9: *L25: cloud microphysical*

AA9: The above correction was applied.

RC10: *L26: development of the clouds, more*

AA10: The above correction was applied.

RC11: *L28: Despite the fact that cloud...*

AA11: The above correction was applied.

RC12: *L31: size distribution of cloud droplets*

AA12: The above correction was applied.

RC13: *L33: cloud lifetime and radiative effects as well as precipitation (e.g....*

AA13: The above suggestion was accepted.

RC14: *L34: McFarquhar*

AA14: The above correction was applied.

RC15: *L35: Three general approaches were used in previous studies of cloud microphysical…*

AA15: The above correction was applied.

RC16: *L36: (e.g. Heymsfield et al., 2011; Craig et al., 2014; Petäjä et al., 2016; Nguyen et al., 2021)*

AA16: Nguyen, C. M., Wolde, M., Battaglia, A., Nichman, L., Bliankinshtein, N., Haimov, S., Bala, K., and Schuettemeyer, D.: Coincident In-situ and Triple-Frequency Radar Airborne Observations in the Arctic, Atmos. Meas. Tech. Discuss. [preprint], https://doi.org/10.5194/amt-2021-148, in review, 2021

The above reference was added.

RC17: *L41: access to individual hydrometeors within a sampling volume. Unfortunately, each of the aforementioned approaches has inherent limitations.*

AA17: The above suggestion was accepted.

RC18: L43: *Data sets that have been obtained from measurements in sub-Arctic clouds are of high value since cloud processes are considered as an important component of climate change in the Arctic…*

AA18: The above suggestion was accepted.

RC19: *L45: The main objective during PaCE…*

AA19: The above correction was applied.

RC20: *L46: In this work, we present a unique dataset – in other places you use 'data sets', please use dataset. Also it's not clear throughout the text which dataset/datasets you are referring to in each sentence e.g. dataset of a single instruments or the whole dataset*

AA20: The above correction was applied. The dataset which we are referring is the whole dataset since there is connection of the cloud microphysical properties with the several meteorological parameters. We will make this clear in the revised manuscript

RC21: *L48: This dataset can be used in studies of cloud microphysics, climate change in the sub-Arctic, and performance evaluation and improvement of existing models, in particular at higher latitudes.*

AA21: The above suggestion was accepted.

RC22: *L49:  In the next section, we provide a description of the sampling location, instrumentation, and the methodology we used for sampling, data processing, and quality control.*

AA22: The above suggestion was accepted.

RC23:*L55: The Sammaltunturi station (67°58´24´´N, 24°06´58´´E) is hosted by the Finnish…*

AA23: The above suggestion was accepted.

RC24: *L58: an excellent location for the monitoring of background…*

AA24: The above correction was applied.

RC25:*L59: The station is about 100 m above the tree canopy line and the vegetation…*

AA25: Tree line is the proper definition in our case. The tree line corresponds" to the edge of the habitat at which trees are capable of growing. Beyond the tree line, trees cannot tolerate the environmental conditions" (Wikipedia). This is exactly the situation we are facing in the Pallas area.

RC26:*L60: (see Lohila et al. 2015).*

AA26: The above suggestion was accepted.

RC27: *L66: The predominant origin of air masses arriving at Sammaltunturi is from…*

AA27: The above correction was applied.

RC28: *L68: The main motivation to perform in-situ cloud measurements at the Sammaltunturi was that the station is occasionally immersed in a cloud.*

AA28: The above suggestion was accepted.

RC29*: L69: measurements was autumn when the horizontal visibility drops bellows…*

AA29: The above suggestion was accepted.

RC30: *L70: Once the preferable time of the year was identified, we started to conduct…*

AA30: The above suggestion was accepted.

RC31*: L71: The "Pallas Cloud Experiments" were, usually, 6-8 weeks…*

AA31: The above correction was applied.

RC32: *L72: occasionally extended to the beginning of December.*

AA32: The above suggestion was accepted.

RC33: *L73: attempt of measuring in situ cloud properties – throughout the text, you sometimes use 'in situ', 'in-situ', 'in-situ', choose one*

AA33: in situ was used throughout the whole revised manuscript.

RC34: *L76: remove (includes two instruments; the cloud and aerosol spectrometer (CAS) and the cloud imaging probe (CIP)) – you will mention this in L113.*

AA34: The above suggestion was accepted.

RC35: *L76: was added.*

AA35: The above suggestion was accepted.

RC36: *L76: In January and February 2012, it was tested for the first time for two short periods…*

AA36: The above suggestion was accepted

RC37:*L94: sensors can be found in Hatakka…*

AA37: The above suggestion was accepted

RC38:*L99: The CAPS was fixed and…*

AA38: The above suggestion was accepted

RC38:*L100: was installed on a rotating platform…*

AA38: The above correction was applied.

RC39: *L101: The CAPS had a total height of 0.6 m above the roof where…*

AA39: The above correction was applied.

RC40*: L102: 0.6 m above…*

AA40: The above correction was applied.

RC41*: L104: was often blocked by freezing of supercooled…*

AA41: The above suggestion was accepted.

RC42: *L105: it was cleaned every hour if occurrence of supercooled water was detected.*

AA42: The above suggestion was accepted.

RC43:*L107: However, even without placing the laminator, the Reynolds number indicated that the…*

AA43: The above suggestion was accepted.

RC44:*L109: were more extensive and the number of cases when the FSSP would have been blocked was significantly reduced.*

AA44: The above suggestion was accepted.

RC45:*L111: documented in Doulgeris et al. (2020).*

AA45: The above correction was applied.

RC46:*L112: include reference to DMT Manuals please in your reference list. e.g. Droplet Measurement Technologies Manual: CAPS operator manual, DOC-0066 Revision F, DMT, Boulder, Colorado, USA, 2011.*

AA46: The above reference was added.

RC47: *L112: in airborne measurements of the microphysical properties in clouds…*

AA47: The above suggestion was accepted.

RC48: *L113: you should add Lachalan-Cope et al. 2016 to the ref list here*

AA48: The above reference was added.

RC49:*L116: supercooled liquid clouds (even for a short time) the sensor was accreting ice.*

AA49: The above correction was applied.

RC50: *L117: significantly shorter than the duration of the campaign. The FSSP-100 was widely used*

AA50: The above correction was applied.

RC51: *L118: 'widely used' but you provide only one reference from 1989?*

AA51:  reference list was updated with Doulgeris et al.2020; Lihavainen et al.2008, Lloyd et al ,2015)

RC52: *L118: CAS and FSSP-100 derive the size of the particle from the intensity of the forward scattered…*

AA52: The above correction was applied.

RC53: *L121: whereby single particles pass through a collimated laser beam and their shadow is projected…*

AA53: The above correction was applied.

RC54: *L122: The count of the particle is dependent on the change in the light intensity of each diode.*

AA54: The above correction was applied.

RC55:*L124: perform calibration at the FMI, on top of manufacturer calibration, to ensure the quality of the collected?processed? data.*

AA55: "Produced" changed to "collected"

RC56: *L125: CAS and FSSP-100, glass beads…*

AA56: The above correction was applied.

RC57: *L128: (Baldor, Reliance, USA), which was working as an aspiration system. - please change everywhere to aspiration system.*

AA57: inhalation was changed to aspiration as it was suggested.

RC58*: L130: employed through FSSP-100 inlet to ensure constant flow…*

AA58: The above suggestion was accepted.

RC58*:L131: was used in each campaign for checks of daily cloud…*

AA59: The above suggestion was accepted.

RC60: *L133: In those years, a necking inside…*

AA60: In this case, "there" corresponds to place (inside the inlet) and not to time (years). "There" will be deleted in the revised manuscript to avoid the misunderstanding.

RC61: *L136: snow or ice could accrete and affect…*

AA61: The above suggestion was accepted.

RC62*:L139: size distribution. The PADS software..*

AA62: The above suggestion was accepted.

RC63*:L140: (DMT Manual, 2011) - This manual should be cited in L112. Here you should put Droplet Measurement Technologies Manual: Particle Analysis and Display System (PADS) Image Probe Data Reference Manual DOC-0201 Rev A-2 PADS 2.5.6, DMT, Boulder, Colorado, USA, 2009.*

*-FSSP manual reference should also be provided where relevant.*

AA63: The above reference was added.

Also FSSP manual reference was added where relevant.

RC64*:L140: provided the number concentration…*

AA64: The above suggestion was accepted.

RC65: *L141: and effective diameter, (ED, μm) - I don't think this parameter is used by anyone. I believe DMT also recommend not to use it since it doesn't have any physical meaning in real clouds. Unless you know any recent studies that used DMTs' ED? Please mention these when you discuss how this dataset can be used.*

AA65: We respect that ED reliability as a parameter could be under discussion for some people, however our aim in not to discuss its reliability as a parameter in real clouds but to provide a complete data set with the derived parameters that CAPS and FSSP are producing. There are plenty of cases that the

effective diameter or radius was used in literature (e.g. Mitchell et al. 2011; Twohy et al 2013; Vivekanandan et al. 2020; Luo et al. 2021). There are also recent manuscripts where particularly DTMs effective diameter or radius was used in literature (e.g. Sorooshian et al. 2018; Calcan et al.2021; Vâjâiac et al. 2021). Thus, we would like to keep the ED in the provided dataset.

RC66: *L143: PADS 2.2?*

AA66: the DMT PACS software (version 2.2.0) was used for the FSSP, it is a predecessor of PADS software.

RC67:L144: remove: we used for cloud measurements, remove the e letter 'are summarized'

AA67: The above correction was applied.

RC68:*L147: In the given data set, only measurements when the station was inside a cloud were used. – I suggest to rephrase: The current dataset contains only in-cloud measurements, when the station was immersed in a cloud.*

AA68: The above suggestion was accepted.

RC69:*L148: format for release and further analysis.*

AA69: The above suggestion was accepted.

RC70: *L149: size distribution measured in both cloud spectrometers.*

AA70: The above correction was applied.

RC71: *L155: During PaCE 2009*

AA71: The above correction was applied.

RC72:  *set to sample each 1 s.*

AA72: The above suggestion was accepted.

RC73: *L158: sampling time was 15 s. For every year, one minute averages…*

AA73: The above correction was applied.

RC74:*L167: horizontal wind direction…*

AA74: The above correction was applied.

RC75: *L183: Measurements of each year were inspected to ensure a good quality of the dataset:*

AA75: The above correction was applied.

RC76: *L184: further analysis cases when one of the cloud probes was partially…*

AA76: The above suggestion was accepted.

RC77: *L185: Then, we used the suggested limitations - what does it mean? Perhaps: applied suggested corrections/filtering due to limitations?*

AA77: "applied the suggested corrections due to" was used instead of "used the suggested limitations" in the revised manuscript.

RC78:*L186: Doulgeris et al. (2020) demonstrated that the CAPS (that was fixed to one direction) showed significant sampling losses…*

AA78: The above suggestion was accepted.

RC79:*L191: PaCE?*

AA79: The above correction was applied.

RC80: *L192: duration are significantly higher. The amount of data in these years is excessive serving as an important source of information for Arctic studies. An overview…*

AA80: The above suggestion was accepted.

RC81: *L193: each campaign when the FSSP-100 and CAPS ground setups were operational.*

AA81: The above correction was applied.

RC82:*L194: In Fig. 6, we show the percentage of the data set for each year in which the Global solar radiation was higher than 0. It was used to estimate the amount of data collected in each campaign in day light.*

AA82: The above suggestion was accepted.

RC83: *L196: In addition, an overview…*

AA83: The above suggestion was accepted.

RC84: *L196: Thus, in Figs. 7-9 - you mention three figures in one sentence, without any further discussion, this should be expanded significantly.*

AA84: Description in figures 7 – 9 will be expanded in the revised manuscript.

RC85: *L198: each campaign and for FSSP-100 and CAS ground setups, respectively.*

AA85: The above correction was applied.

RC86: *L223: Such semi long observations are difficult to obtain in similar environments due to current lack of instrumentation - this sentence should probably be elaborated further, instead of text repetitions like in lines 224-226.*

AA86: This aspect will be elaborated in the discussion of the revised manuscript.

"..instrumentation which will also allow continuous unattended operation at temperature below 0 ºC. Cloud droplet spectrometers with surface installation had been identified as a potential method for continuous cloud in-situ measurements (Wandinger et al., 2018). Thus, due to the increased demand for long term continuous ground based in-situ cloud measurements, we provide a data set of in situ cloud measurements in a harsh sub arctic environment"

RC87: *L224: remove: (size distribution as a measured parameter and additionally as derived parameters the number concentration, effective diameter, median volume diameter and the liquid water content)*

AA87: The above suggestion was accepted.

RC88: *L226: remove: (temperature, dew point temperature, humidity, pressure, wind speed, wind direction, (global solar) sun radiation and visibility)*

AA88: The above suggestion was accepted.

RC89: *L230: of observations per year and operators' experience running the ground-based…*

AA89: The above suggestion was accepted.

RC90: *L232: However, this data set provides a helpful contribution to cloud microphysics processes on shorter timescales. Furthermore, it can be used as complementary in model development.  -  Please provide examples how this data can be used? Which models, could you provide examples of models where it can be used?*

AA90: In atmosphere, microphysics corresponds to the physical and chemical processes occurring at the scale of individual cloud and precipitation particles, or hydrometeors (sub-micron to several centimeters). In this work, we make available a dataset of microphysical cloud properties along several meteorological parameters in order to represent cloud microphysics processes in several low level cloud cases. The representation of cloud microphysics is significant for large eddy simulation models (LES) (Morrison et al. 2020). Also, microphysical processes can strongly influence cloud-climate feedbacks in global climate models (Bodas – Salcedo et al. 2019).

RC91: *Figure 1 caption: Map of Finland showing the location of the field station, and (b) map of the wider Pallas area showing the location of the Sammaltunturi station…*

AA91: The above suggestion was accepted.

RC92: *Figure 6 caption: The percentage of the global solar radiation that was higher than 0 during each campaign when the FSSP-100 and CAS ground setups were operational.*

AA92: The above correction was applied.

RC93: *Figure 8: Why you use ED but no MVD figure? Did you get any images at all in the Cloud Imaging Probe that you can present here? If CIP did record images, I'd recommend to upload the raw files dataset as well. Otherwise, there should be an explanation why images are not included.*

AA93: The reason we chose not to include MVD figure was because the values of MVD were quite close to the values of ED. However, MVD figure will be added in the revised manuscript as the reviewer suggested.

When the CIP was operational, we also got the CIP images. However, we did not include the raw images in the data set for two reasons, First, there were in binary format. To read them, we used a proprietary image analysis software that was provided by DMT. Secondly, the upper limit of the open data repository is 10GB which was not enough to include the CIP raw images which were approximately 0,5 GB per case/day. However, RAW CIP images could be provided on demand by authors.

**References**

Bodas-Salcedo, A., Mulcahy, J. P., Andrews, T., Williams, K. D., Ringer, M. A., Field, P. R., & Elsaesser, G. S. (2019). Strong dependence of atmospheric feedbacks on mixed-phase microhysics and aerosol-cloud interactions. Journal of Advances in Modeling Earth Systems, 11, 1735–1758. https://doi.org/10.1029/2019MS001688.

Calcan, A., Stefan S., Vâjâiac, S. N. and Moacă, D.-E. (2021). Airborne measurements in different clouds. INCAS BULLETIN. 13. 19-28. 10.13111/2066-8201.2021.13.1.3.

Doulgeris, K.-M., Komppula, M., Romakkaniemi, S., Hyvärinen, A.-P., Kerminen, V.-M., and Brus, D.: In situ cloud ground-based measurements in the Finnish sub-Arctic: intercomparison of three cloud spectrometer setups, Atmos. Meas. Tech., 13, 5129–5147, https://doi.org/10.5194/amt-13-5129-2020, 2020.

Forward Scattering Spectrometer Probe, PSM model FSSP-100, 0.5-47 µm operating and servicing manual, PMS, Boulder, Colorado, USA, S/N E01284-0494-156.

Lachlan-Cope, T., Listowski, C., and O'Shea, S.: The microphysics of clouds over the Antarctic Peninsula – Part 1: Observations, Atmos. Chem. Phys., 16, 15605–15617, https://doi.org/10.5194/acp-16-15605-2016, 2016.

Lihavainen, H., Kerminen, V.-M., Komppula, M., Hyvärinen, A.-P., Laakia, J., Saarikoski, S., Makkonen, U., Kivekäs, N., Hillamo, R., Kulmala, M., and Viisanen, Y.: Measurements of the relation between aerosol properties and microphysics and chemistry of low-level liquid water clouds in Northern Finland, Atmos. Chem. Phys., 8, 6925–6938, https://doi.org/10.5194/acp-8-6925-2008, 2008.

Lloyd, G., Choularton, T. W., Bower, K. N., Gallagher, M. W., Connolly, P. J., Flynn, M., Farrington, R., Crosier, J., Schlenczek, O., Fugal, J., and Henneberger, J.: The origins of ice crystals measured in mixed-phase clouds at the high-alpine site Jungfraujoch, Atmos. Chem. Phys., 15, 12953–12969, https://doi.org/10.5194/acp-15-12953-2015, 2015.

Luo, Q.; Yi, B.; Bi, L. Sensitivity of Mixed-Phase Cloud Optical Properties to Cloud Particle Model and Microphysical Factors at Wavelengths from 0.2 to 100 µm. *Remote Sens.* **2021**, *13*, 2330. https://doi.org/10.3390/rs13122330

Measurement Technologies Manual: Particle Analysis and Display System (PADS) Image Probe Data Reference Manual DOC-0201 Rev A-2 PADS 2.5.6, DMT, Boulder, Colorado, USA, 2009.

Mitchell, D. L., Lawson, R. P., and Baker, B.: Understanding effective diameter and its application to terrestrial radiation in ice clouds, Atmos. Chem. Phys., 11, 3417–3429, https://doi.org/10.5194/acp-11-3417-2011, 2011.

Morrison, H., van Lier-Walqui, M., Fridlind, A. M., Grabowski, W. W., Harrington, J. Y., Hoose, C., et al. (2020). Confronting the challenge of modeling cloud and precipitation microphysics. Journal of Advances in Modeling Earth Systems, 12, e2019MS001689. https://doi.org/ 10.1029/2019MS001689

Sorooshian, A., MacDonald, A., Dadashazar, H. *et al.* A multi-year data set on aerosol-cloud-precipitation-meteorology interactions for marine stratocumulus clouds. *Sci Data* **5,** 180026 (2018). https://doi.org/10.1038/sdata.2018.26

Twohy, C. H., Anderson, J. R., Toohey, D. W., Andrejczuk, M., Adams, A., Lytle, M., George, R. C., Wood, R., Saide, P., Spak, S., Zuidema, P., and Leon, D.: Impacts of aerosol particles on the microphysical and radiative properties of stratocumulus clouds over the southeast Pacific Ocean, Atmos. Chem. Phys., 13, 2541–2562, https://doi.org/10.5194/acp-13-2541-2013, 2013.

Vâjâiac, S. N., Calcan, A., David, R. O., Moacă, D.-E., Iorga, G., Storelvmo, T., Vulturescu, V., and Filip, V.: Post-flight analysis of detailed size distributions of warm cloud droplets, as determined in situ by cloud and aerosol spectrometers, Atmos. Meas. Tech. Discuss. [preprint], https://doi.org/10.5194/amt-2021-185, accepted, 2021.

Vivekanandan, J., Ghate, V. P., Jensen, J. B., Ellis, S. M., & Schwartz, M. C. (2020). A Technique for Estimating Liquid Droplet Diameter and Liquid Water Content in Stratocumulus Clouds Using Radar and Lidar Measurements, Journal of Atmospheric and Oceanic Technology, 37(11), 2145-2161.

---

## Author Comment (AC2)

We sincerely thank the reviewer #2 for carefully reading of our manuscript, for his review and constructive comments. We have reviewed the comments and have revised the manuscript accordingly. Our response is given in a point-by-point manner below. Reviewer comment (RC) Authors answer (AA).

*There is a lot of value in these datasets and descriptions, and the robust and detailed comments from reviewer#1 are very good and improve the manuscript. My comments:*

We sincerely thank the reviewer for highlighting the value in these datasets. We indeed agree that comments from reviewer 1 were significant and improved the manuscript.

**Major comments:**

*RC1: The data are described and presented as if the data are perfect. There is no description or estimate of error or uncertainty in the data, no discussion of data processing, limited discussion of calibration, and no discussion of potential sampling artefacts aside from identifying periods where inlets are frozen. This is quite challanging to do, but if the idea is to put this dataset out into the public domain as being representative ambient data, then it needs a complete description.*

AA1:

We agree with the reviewer that an extended description was needed. Thus, a discussion about possible uncertainties in the dataset, calibration and data processing was added in the revised manuscript. Table 1 of the manuscript already presented some of the uncertainties that have been found in literature. On top, the following text was added in line 150:

**"** The major sources of uncertainties of the cloud spectrometers can be coincidence, dead time losses and changing velocity ratio (Guyot et al.,2015). The uncertainty of estimation of sizing at the cloud spectrometers was as 20% and of the number concentration was as 16% (Baumgardner, 1983; Dye and Baumgardner, 1984; Baumgardner et al., 2017). According to Lance (2012), it was observed that for CAS at ambient droplet concentrations of 500 cm$^{-3}$ there was 27 % undercounting and a 20 %–30 % oversizing bias. In our case, during PaCE campaigns the droplet number concentration values we monitored were in the majority of cases less than 300 cm$^{-3}$. These number concentration values lead us not to take coincidence, dead-time losses, and VAR uncertainties into consideration in this analysis. LWC has a significant uncertainty of 40% (DMT manual 2011). The FSSP derived ED and LWC had an uncertainty of 3 µm and 30 % in mixed-phase clouds (Febvre et al. ,2012)."

The discussion regarding calibration was extended. The following text was added in line 129:

"Cloud spectrometers (in our case CAS and FSSP-100) are calibrated for size measurements but not for number concentration measurements. The instruments faced extreme conditions during the whole campaign, in terms of frequent changes in wind direction, windspeed and sub-zero temperatures. Despite the calibration procedures we should always keep in mind that extreme meteorological conditions could possibly lead to unexpected performance."

The data limitation and processing were analyzed in detail in Doulgeris et al, 2020. To give more details also in the current manuscript the discussion regarding data processing and possible artifacts was extended. The following text was added in line 199:

"..or fully blocked. Partially or fully blocked probes were also visible in raw data. To detect blocked probes, Nc was carefully investigated for the whole data set. When a sudden decrease just before a sudden

increase in droplet number concentration was occurring, we had a clear sign of probe inlet freezing. This behavior was observed due to the opening of the probe inlet becoming smaller (from the accumulation of snow/ ice) and resulted in a raised probe air speed. During data evaluation we considered that the probe air speed was constant. This abnormality in the Nc was happening due to the underestimation of the probe air speed."

The following text was added in line 206: " … the wind direction since it was not sampling isokinetically".

**Minor comments:**

RC2: *It can be very difficult to obtain instrumenet manuals. Sometimes urls change and links become defunct. Is there a way in which the relavent manuals can be provided with th manuscript, or permament links established?*

AA2:

We agree with the reviewer that it is very difficult to obtain instrument manuals. The manuals were provided to us along with the instruments purchase and we don't have the right to publish them. Their online availability is dependent on the instruments manufacturer. However, we provide all the existing manuals that are available online in our reference list (DMT Manual, 2009; DMT Manual, 2011).

RC3: *With time there can be changes to firmware used during data aquisition, and sometimes this can affect. Were there any changes in Firmware between projects? The software and firmware used for data aquisition/processing should at least be documented in some form of table.*

AA3:

We used one version of the PADS software for the CAPS and one version of PACS software for the FSSP during all PaCEs. The above info was clarified in line 142 of the manuscript "The ground-based in-situ cloud measurements provided the cloud and precipitation size distribution. On top, the PADS 2.5.6 software that was used for the data acquisition of CAPS measurements (DMT Manual, 2011), derived the number concentration (Nc, cm$^{-3}$), liquid water content (LWC, g cm$^{-3}$), median volume diameter (MVD, µm) and effective diameter, (ED, µm). For the FSSP100, Nc, LWC, MVD and ED were also derived using the same equations (Doulgeris et al., 2019), since we have used an older software for data acquisition (PACS 2.2, DMT). "

Regarding the data analysis we clarified in code availability section that "Software developed to process and display the data from the cloud ground base spectrometers are not publicly available and leverages licensed data analysis software (MATLAB). This software contains intellectual property that is not meant for public dissemination."

**References**

Baumgardner, D.: An analysis and comparison of five water droplet measuring instruments, J. Appl. Meteorol., 22, 891–910, https://doi.org/10.1175/1520- 0450(1983)0222.0.CO;2, 1983.

Baumgardner, D., Abel, S. J., Axisa, D., Cotton, R., Crosier, J., Field, P., Gurganus, C., Heymsfield, A., Korolev, A., Krämer, M., Lawson, P., McFarquhar, G., Ulanowski, Z., and Um, J.: Cloud Ice Properties: In Situ Measurement Challenges, Meteor. Mon., 58, 9.1–9.23, https://doi.org/10.1175/AMSMONOGRAPHS-D16-0011.1, 2017.

Droplet Measurement Technologies Manual: CAPS operator manual, DOC-0066 Revision F, DMT, Boulder, Colorado, USA, 2011.

Dye, J. E. and Baumgardner, D.: Evaluation of the forward scattering spectrometer probe, I – Electronic and optical studies, J. Atmos. Ocean. Technol., 1, 329–344, https://doi.org/10.1175/1520-0426(1984)0012.0.CO;2, 1984.

Febvre, G., Gayet, J.-F., Shcherbakov, V., Gourbeyre, C., and Jourdan, O.: Some effects of ice crystals on the FSSP measurements in mixed phase clouds, Atmos. Chem. Phys., 12, 8963–8977, https://doi.org/10.5194/acp-12-8963-2012, 2012.

Forward Scattering Spectrometer Probe, PSM model FSSP-100, 0.5-47 µm operating and servicing manual, PMS, Boulder, Colorado, USA, S/N E01284-0494-156.

Lance, S.: Coincidence Errors in a Cloud Droplet Probe (CDP) and a Cloud and Aerosol Spectrometer (CAS), and the Improved Performance of a Modified CDP, https://doi.org/10.1175/JTECH-D11-00208.1, 2012.

Measurement Technologies Manual: Particle Analysis and Display System (PADS) Image Probe Data Reference Manual DOC-0201 Rev A-2 PADS 2.5.6, DMT, Boulder, Colorado, USA, 2009